# Evaluation of Landslide Susceptibility Based on CF-SVM in Nujiang Prefecture

**DOI:** 10.3390/ijerph192114248

**Published:** 2022-10-31

**Authors:** Yimin Li, Xuanlun Deng, Peikun Ji, Yiming Yang, Wenxue Jiang, Zhifang Zhao

**Affiliations:** 1College of Earth Sciences, Yunnan University, Kunming 650500, China; 2Engineering Research Center of Domestic High-Resolution Satellite Remote Sensing Geology for Universities of Yunnan Province, Kunming 650500, China

**Keywords:** GIS, landslide susceptibility, certainty factor, support vector machine, model coupling, Nujiang Prefecture

## Abstract

At present, landslide susceptibility assessment (LSA) based on landslide characteristics in different areas is an effective measure for landslide management. Nujiang Prefecture in China has steep mountain slopes, a large amount of water and loose soil, and frequent landslide disasters, which have caused a large number of casualties and economic losses. This paper aims to understand the characteristics and formation mechanism of regional landslides through the evaluation of landslide susceptibility so as to provide relevant references and suggestions for spatial planning and disaster prevention and mitigation in Nujiang Prefecture. Based on the grid cell, this study selected 10 parameters, namely elevation, slope, aspect, lithology, proximity to faults, proximity to road, proximity to rivers, normalized difference vegetation index (NDVI), land-use type, and precipitation. Support vector machine (SVM), certainty factor method (CF), and deterministic coefficient method–support vector machine (CF-SVM) were used to evaluate the landslide susceptibility in Nujiang Prefecture. According to these three models, the study area was divided into five landslide susceptibility grades, including extremely high susceptibility, high susceptibility, moderate susceptibility, low susceptibility, and very low susceptibility. Receiver operating characteristic curve (ROC) was applied to verify the accuracy of the model. The results showed that CF model (ROC = 0.865), SVM model (ROC = 0.892), CF-SVM model (ROC = 0.925), and CF-SVM model showed better performance. Therefore, CF-SVM model results were selected for analysis. The study found that the characteristics of high and extremely high landslide-prone areas in Nujiang Prefecture have the following characteristics: intense human activities, large density of buildings and arable land, rich water resources, good economic development, perfect transportation facilities, and complex topography and landform. In addition, there is a finding inconsistent with our common sense that the distribution of landslide disasters in the study area does not decrease with the increase of NDVI value. This is because the Nujiang River basin is a high mountain canyon area with low rock strength, barren soil, and underdeveloped vegetation and root system. In an area with large slope, the probability of landslide disaster will increase with the increase of NDVI. The CF-SVM coupling model adopted in this study is a good first attempt in the study of landslide hazard susceptibility in Nujiang Prefecture.

## 1. Introduction

Landslide refers to the natural phenomenon in which the soil or rock mass on the slope slides down the slope as a whole or in a scattered manner along a certain weak surface or zone, caused by various natural processes and human activities under the action of gravity [1,2,3,4,5,6]. Whether caused by natural factors or human activities, landslides cause a great deal of economic losses and loss of life every year [7,8,9]. Extreme natural events such as landslides cannot be foreseen, but their risk can be reduced by taking precautions and early warning measures. Landslide susceptibility assessment aims to predict and analyze the spatial and temporal distribution and occurrence probability of landslide hazard [10]. In recent decades, the most appropriate way to prevent landslide events has been to conduct spatial assessment of landslide susceptibility [11,12]. Currently, a series of studies around the world has assessed landslide susceptibility in the context of multiple influencing factors. The results can provide an important decision-making basis for landslide hazard risk management, territorial space planning and layout, and landslide monitoring [13].

Major advances in computing power, remote sensing, and geographic information systems (GIS) have facilitated the development of landslide susceptibility maps. Various landslide susceptibility mapping models and methods have been proposed, including: (1) knowledge-based approaches such as analytic hierarchy process [14] and expert scoring method [15]; (2) data-driven approaches such as information content method [16], frequency ratio [17], certainty factor [18], index of entropy [19], and logistic regression analysis [20]; and (3) machine-learning methods (ML), such as decision tree [21,22], artificial neural network (ANN) [23], support vector machine (SVM) [24], and random forest [25]. The combination of GIS with data-driven methods or machine learning methods (ML) has been widely used for landslide susceptibility assessment using spatial and non-spatial data [17,23,24,25]. For example, Cheng, Y. et al. [26] used analytic hierarchy process to study highway tunnel landslides. Chen, F. et al. [16] used the information model for spatial susceptibility prediction and analysis of landslides. Ali, S. et al. [27] used GIS technology to draw the landslide susceptibility map along the China–Pakistan Economic Corridor (Karakoram Highway). Klose et al. [28] assessed landslide susceptibility in northwestern Central Europe using a landslide geographic information database and bivariate statistical analysis, while Niu, R.Q. et al. [29] performed a susceptibility evaluation using ML, and Wu et al. [30] used decision tree and support vector machine to evaluate the landslide susceptibility of Guojiaba Town in the Three Gorges Reservoir Area, China. Most of the above studies applied multiple single models to landslide susceptibility and did not consider coupling studies of different types of models.

A total of 6181 geological disasters occurred in China, including 4220 landslides, accounting for 68.27 percent of the total according to the 2019 National Geological Disaster Bulletin. Thus, we can see the seriousness of landslide geological disasters. Nujiang Prefecture in China is a key area for the prevention and control of landslide geological disasters. Because of its complex terrain, unique climate, and fragile ecological environment, landslides occur frequently, seriously affecting the local economic and social development [31,32]. At present, it is urgent to evaluate the susceptibility of landslide disasters in Nujiang Prefecture. Landslide susceptibility assessment (LSA) is a typical non-engineering measure and an effective method to mitigate, prevent, and manage landslide damage [8,33,34,35,36,37,38]. In this research area, some experts have carried out landslide disaster research. For example, He, J. et al. [39] analyzed landslide inducing factors in Lushui County, Nujiang Prefecture, from the aspects of stratigraphic lithology, geological structure, earthquake, and rainfall. Tan, S.C. et al. [40] designed and established the spatial database for meteorological forecasting and the early warning of slope geological disasters, which solved the limitation of the traditional database, which could not effectively manage spatial data. Hu, J. et al. [41] optimized the critical rainfall, critical rain intensity and prone zoning of precipitation-type geological disasters in Yunnan Province. Li, Y.M. et al. [42] used the CF to analyze the susceptibility of slope geological disasters, analyzed the influence degree of each factor on the occurrence of slope geological disasters through the susceptibility index, and drew the zoning map of slope geological disaster susceptibility. Yao, Y.W. et al. [43] used the distribution of geological disaster points to analyze the characteristics of geological disasters in Lanping County, Nujiang Prefecture, and put forward countermeasures for disaster prevention and control. Although some people have studied the landslide disaster in Nujiang Prefecture, there are few studies on its susceptibility carried out by using models. In the global scale, few people use mathematical statistics model and machine learning model coupling to study the landslide susceptibility.

At present, scholars use a variety of models to study landslide susceptibility. Among these models, CF and SVM models have been widely used and praised. Theoretically, CF is a mathematical statistical model, which has advantages in calculating CF values of various levels of influence factors. SVM is a quantitative and objective model obtained by various numerical calculations, which has great advantages in the study of small sample data.

The research shows that the coupling model can synthesize the advantages of each model and make up for the shortcomings of each model. In terms of evaluation accuracy and success rate, the model has obvious advantages over the single model. The hybrid application of the model can significantly improve the accuracy and reliability of the results. Therefore, this study selected two single models (CF model and SVM model) and one mixed model (CF-SVM) to assess landslide susceptibility through the analysis of landslide data in Nujiang Prefecture. The advantages and disadvantages of the deterministic coefficient model and support vector machine are complementary, which effectively improves the prediction accuracy of the model. This study provides a reference for the selection of regional landslide susceptibility evaluation model under similar geological conditions. It will help protect people’s lives and property, reduce disaster losses, and improve the efficiency of disaster prevention and mitigation. It can also provide scientific basis for government departments to formulate disaster prevention and mitigation measures, which has important application value.

## 2. Materials and Methods

### 2.1. Overview of the Study Area and Data Sources

#### 2.1.1. Overview of the Study Area

In this study, Nujiang Prefecture was selected as the study area. Nujiang Prefecture is located in the longitudinal ridge and valley area of the Hengduan Mountains in the northwest of Yunnan Province. It is located at 25°33′–28°23′ E and 98°09′–99°39′ N. It includes the Lushui, Fugong, Gongshan, and Lanping Counties. The total area of the four counties is 14,703 km^2^ (Figure 1). Nujiang Prefecture is located in a unique plateau and mountainous environment, which is a typical deep-cutting zone of alpine valleys. More than 40 mountain peaks exceed 4000 m. It includes four mountain ranges and three rivers: Lika Mountain, Dulong River, Gaoligong Mountain, Nujiang River, Biluo Snow Mountain, Lancang River, and Yunling Mountain. The three rivers slope across the entire territory from north to south, forming a typical alpine and canyon landform. Due to plate collision and subduction, a series of deep fault zones were formed in Nujiang Prefecture. Affected by erosion and gravity, the rock mass is broken, loose materials are piled up, and the steep slope reclamation phenomenon in Nujiang Prefecture is obvious, leading to serious soil erosion and frequent geological disasters. Nujiang Prefecture has a subtropical mountain monsoon climate, and its valley areas present subtropical humid climate characteristics. The mountain ranges are alternately affected by the Qinghai–Tibet Plateau and the Bay of Bengal air currents, coupled with the large topographical and vertical climate differences.

#### 2.1.2. Data Sources

Table 1 lists the landslide conditioning factors used in this study, together with their sources and scales. Among them, the data of historical disaster points were provided by the project team employed in this study, with a total of 561 landslide disaster points.

### 2.2. Models

The deterministic coefficient model can determine landslide susceptibility based on the relationship between past landslide points and hazard-inducing factors, that is, quantitatively reflecting the susceptibility interval of a certain hazard-inducing factor. However, it is impossible to reflect the contribution of this factor in landslide occurrence as a whole. In contrast, the SVM model is not prone to overfitting in case of limited samples, yields good performance in the classification process, and can characterize the degree of contribution of the evaluation factors. Therefore, coupling the two models allows utilizing the advantages of both; that is, the deterministic coefficient model is used for calculating the susceptibility of different classes between factors for classification of landslide data and non-landslide data, and the SVM model is used for training and prediction to improve the evaluation accuracy.

#### 2.2.1. Deterministic Coefficient Model

The certainty factor method (CF) is a probability function belonging to the category of bivariate statistical analysis and can be used to analyze the susceptibility of disaster events according to various factors.
(1)$$\mathrm{CF}\;=\left\{\begin{array}{c} \frac{\mathrm{PPa}-\mathrm{PPs}}{\mathrm{PPa}\;\left(1-\mathrm{PPs}\right)}\;\;\;,\;\mathrm{PPa}>\mathrm{PPs}\\ \frac{\mathrm{PPa}-\mathrm{PPs}}{\mathrm{PPs}\;\left(1-\mathrm{PPa}\right)}\;\;\;,\;\;\mathrm{PPa}<\mathrm{PPs} \end{array}\right.$$

PPa is the probability of geological disasters occurring in the evaluation factor A, and when applied, it is the ratio of the number of geological disasters existing in the evaluation factor A to the area of the factor A. PPs is the prior probability of geological disasters occurring in the whole study area, that is, the ratio of the number of geological disasters in the whole study area to the total study area. The variation range of CF is [−1, 1], and the positive value represents a high certainty of geological disaster occurrence. The negative value represents a low certainty of geological disaster occurrence. When the calculated result is close to 0, it means that the factor cannot determine whether the given area is prone to geological disasters.

#### 2.2.2. SVM Model

Support vector machine (SVM) is a binary classification model, which is a classification prediction model developed on the basis of statistical principles [9]. This method is widely used in various fields. SVM is more reasonable and effective than other learning methods in solving small sample, high dimensional, and non-linear problems. The basic principle is to find an optimal hyperplane, which can not only correctly divide the two types of sample points but also maximize the geometric interval from the nearest sample point to the plane. SVM is suitable for small samples and nonlinear and high-dimensional space problems, which can highlight its unique advantages and can be combined with other machine learning to jointly analyze problems. In this study, the training set is set as $T=\left\{\left(x_{1},y_{1}\right)\left(x_{2,}y_{2}\right),\ldots ,\left(x_{n},y_{n}\right)\right\}$, where x is the input vector, and $x_{1}\sim x_{n}$ represents elevation, aspect, slope, lithology, proximity to faults, proximity to road, proximity to river, NDVI, precipitation, and land-use type, respectively. In the formula, $y\in \left(0,\;1\right),$ 1 and 0 denote landslide and non-landslide, respectively. The goal of SVM classification is to find an optimal separating hyperplane that can be distinguished between landslides and non-landslides in the above training set. The equation of the separating hyperplane is: w∗x+b=0, and w is called the vector, and b is called the intercept. The prediction accuracy of SVM depends on the choice of kernel function. There are four types of kernel functions commonly used: linear kernel function, polynomial kernel function, radial basis kernel function, and Sigmoid kernel function. RBF is widely used in landslide susceptibility prediction. Its advantages are its fewer parameters, strong flexibility, and good performance. Therefore, this study adopts RBF kernel function to build support vector machine model, as shown in Formula (2).
(2)$$K\;\left(x_{i}x\right)=\exp{\left(-\gamma \left|\left|x_{i}-x\right|\right|\right)}^{2}$$
where x is the input vector, and γ is the gamma parameter.

#### 2.2.3. CF-SVM Model

When selecting training samples, most studies often take a certain number of landslide data as samples for training and the rest as test samples [34]. These research methods only consider the contribution of each evaluation factor to landslide formation, that is, only analyze the corresponding relationship between evaluation factor and sample without analyzing the stable slope (negative sample), so the selection of sample is often one-sided, and it is difficult to give an in-depth explanation of the mechanism of landslide formation. On the basis of summarizing relevant research, this study adopts the method of positive sample and negative sample for sample selection. The specific process is as follows: Firstly, the CF model is used to partition landslide susceptibility, and then, non-landslide samples are selected from the very-low- and low-susceptibility areas, making the overall sample more reasonable and more authoritative.

The input variables of the model were 561 landslide and 561 non-landslide raster cells, among which landslide raster cells were the known landslide cataloguing information above, and non-landslide cells were mainly collected and acquired in the very-low-susceptibility area and the low-susceptibility area in the whole Nujiang Prefecture in a random way. After that, the above 561 landslide and 561 non-landslide grid cells were randomly divided into two parts, 70% of which were used for SVM model training and 30% of which were used for SVM model testing. By applying the above trained and tested SVM model to the CF values of 11 evaluation factors, the spatial distribution of landslide susceptibility in Nujiang Prefecture could be obtained.

### 2.3. Selection and Grading of Evaluation Factors and Sampling Strategy

#### 2.3.1. Selection of Evaluation Factors

The influencing factors of geological disasters were divided into two categories: internal leading factors and external environmental triggering factors [44]. In this study, based on the collected historical disaster, relevant literature, geological conditions of the study area, landslide formation conditions, and development characteristics, the elevation, aspect, slope, proximity to river, lithology, proximity to faults, normalized difference vegetation index (NDVI), proximity to roads, land-use type, and precipitation were used as evaluation factors to construct an evaluation index system for landslide susceptibility assessment.

#### 2.3.2. Correlation Analysis of Evaluation Factors

When there are multiple collinearities between the evaluation factors, the model becomes complicated, and the prediction accuracy of the model decreases. To avoid this, SPSS software was used to analyze the correlation of each evaluation factor. If the absolute value of the correlation coefficient is greater than 0.3, it means that there is a strong correlation between the factors; otherwise, the correlation is weak. Results of the correlation analysis are shown in Table 2. Only the soil type factor exceeded 0.3, and the absolute values of the correlation coefficients among the other evaluation factors were all less than 0.3, indicating that the correlation between the factors except for the soil type is weak and can be used for landslide susceptibility analysis.

#### 2.3.3. Grading of Evaluation Factors

The evaluation factors were classified according to the geological environment of the study area and the spatial distribution characteristics of landslides. The evaluation factors were divided into two types: continuous and discrete. The CF values of each factor is shown in Table 3,the density of disaster point at different levels of each factor is shown in Figure 2, the single-factor grading diagram is shown in Figure 3.

##### Land-Use Type

Different land-use patterns have different impacts on geological disasters. Unplanned land-use patterns destroy the natural environment and aggravate the occurrence of geological disasters.Based on the 2017 land-use data and Google Earth images from 2020, the land-use data-type distribution map of Nujiang in 2020 was obtained in this study by visual interpretation and correction according to first-level classification standards (Figure 2a). By using the deterministic coefficient model, CF values of unused land, grassland, construction land, cultivated land, forest land, and water bodies were calculated (Table 3).The density of disaster point at different levels of each factor is shown in Figure 3a. 

##### Elevation

Although elevation does not directly affect the occurrence of landslides [32], different elevations lead to different factors, such as rainfall, temperature, soil type, vegetation type, and intensity of human activities. Nujiang Prefecture has high mountains, deep valleys, steep slopes, rapid waters, and complex topography and landforms with an elevation difference of more than 4000 m. According to different vertical climatic zones, ArcGIS was used to classify the elevations (Figure 2b), and the CF values of different elevation zones were calculated. It can be seen from Table 2 that when the elevation is less than 1900 m, the certainty coefficient value is close to 1, indicating that within this range, landslide disasters are extremely prone to occur. Superimposing the elevation classification map with the vector of residential areas in Nujiang Prefecture revealed that areas with an elevation of less than 1900 m are densely distributed, human engineering activities such as steep slope reclamation and slope excavation are more frequent, and the damage to the original natural environment is more serious. Most of the water systems are distributed in areas with lower altitudes, which are more likely to cause landslide disasters. In contrast, when the elevation is more than 1900 m, as the altitude increases, the CF value and the possibility of geological disasters gradually decrease.The density of disaster point at different levels of each factor is shown in Figure 3b.

##### Slope

Slope is a key factor affecting the stability of an area. Areas with large slopes are more likely to witness landslides, while areas with flat terrains and small slopes are less likely to have geological disasters. The study area has high mountains and steep slopes, with a slope range of 0.35–88.38°, which was divided into six grades: 0–10°, 10–20°, 20–30°, 30–40°, 40–50°, and >50° (Figure 2c). The CF model was used to calculate the certainty coefficient of each slope grade, and the relationship between the slope and the occurrence of landslide disasters in the study area was analyzed. The results are shown in Table 3. The certainty coefficient of slope within 10–30° is large, indicating the higher possibility of geological disasters in this area.The density of disaster point at different levels of each factor is shown in Figure 3c. 

##### Aspect

Sunny slopes and shaded slopes receive different solar radiation intensities, which affect the vegetation growth, vegetation types, rainfall, and soil moisture [41,42,43], thereby acting as one of the evaluation factors of landslide disasters. The text was based on ArcGIS10.7 and Nujiang Prefecture DEM data, and the aspect raster map was calculated according to a previously defined method [45,46] (Figure 2d). The aspect was divided into plane, north, northeast, east, southeast, south, southwest, west, and northwest. For the nine levels, the CF value of each slope direction classification area was calculated using the deterministic coefficient model, and the results are shown in Table 3.The density of disaster point at different levels of each factor is shown in Figure 3d.

##### Proximity to Rivers

Rivers are an important factor in the development of geological disasters. Deforestation in the slopes and erosion at the river banks result in an increase in the empty area. In addition, the water flow increases the moisture content of the soil and increases the weight and softens the rock and soil, thereby reducing the stability of the slope and increasing the probability of geological disasters. In this study, the data for the Nujiang River system were extracted using the hydrological analysis function in the ArcGIS software, and the multi-ring buffer area was used in the field analysis to establish the river buffer area at 200 m intervals. The buffer area was divided into seven intervals by using the CF model (Figure 2e), and the relationship between geological hazard susceptibility and distance to rivers was analyzed (Table 3). The results showed that the greater the distance from the rivers, the smaller the CF value. The CF value is greater than 0 for distances of 0–600 m, indicating that the distance from the water system plays a vital role in the occurrence of geological disasters. When the distance from the water system is greater than 600 m, the CF value is less than 0, which shows that the influence of the water system on geological disasters is relatively weak at such distances.The density of disaster point at different levels of each factor is shown in Figure 3e.

##### Lithology

Different rock and soil bodies have different lithologies: their structural, physical, and chemical properties and their ability to resist erosion and weathering vary. Therefore, the probability of occurrence of geological disasters also varies. In this study, stratigraphic groups were used as the basic unit to divide the rock and soil mass in Nujiang Prefecture into four groups: soft rock, hard rock, harder rock, and loose rock. The vector data were converted to raster data by using ArcGIS to obtain the Nujiang State lithology classification map shown in Figure 2f. Next, using the CF model, the influence of stratum lithology on the occurrence and development of geological disasters was quantitatively analyzed. The density of disaster point at different levels of each factor is shown in Figure 3f.The results (Table 3) revealed that the harder rock group has the largest certainty coefficient value, indicating that geological disasters are prone to occur in soft and hard interbedded rock formations, such as quartz sandstone, sandstone, shale-intercalated limestone, slate-intercalated basalt, and limestone-interbedded slate, because soft rock interbeds with hard rock, and soft rock becomes a natural sliding bed, creating favorable conditions for the occurrence of geological disasters, while loose rocks (such as sandy clay and sandy gravel) have poor stability and cannot form slopes with large slopes. Hard rocks are resistant to weathering and are not easily eroded; thus, the slopes have good stability and are not prone to geological disasters.

##### Proximity to Faults

Fault structure is one of the indispensable evaluation factors. During the formation of a fault zone, the rock and soil get divided, destroying the integrity and continuity of the rock formation and affecting the stability of the slope. Broken rock mass and loose deposits can easily lead to geological disasters. The faults in Nujiang Prefecture are clustered; the main faults are distributed from north to south, and the other major faults are mainly located along the north-south direction, with less distribution in the east-west direction. The fault zones in the region are densely distributed and structurally developed. In this study, the multi-ring buffer tool was used to divide the intervals with the catastrophic point ratio and the point density curve mutation points in each grading interval as a reference(Figure 3g), a fault buffer with a distance of 400 m was established (Figure 2g), and the certainty coefficient of each interval was calculated (Table 3). The results showed that the greater the distance from the fault zone, the smaller the certainty coefficient. The CF value is larger in the interval of 0–800 m, indicating that geological disasters are more likely to occur. When the distance from the fault zone exceeds 2000 m, the CF value is less than 0. Thus, the fault structure has little influence on the occurrence of geological disasters at such increased distances.

##### Proximity to Road

The transportation network in Nujiang Prefecture is developing rapidly with the construction of numerous bridges and tunnels. The excavation of slopes and blasting in engineering construction activities disturb the rock and soil mass AND destroy the stability of the slope, and the rock becomes loose and fragile, thus causing geological disasters. According to the road distribution map of Nujiang Prefecture, a buffer zone with a 200 m radius was established (Figure 2h); the deterministic coefficient values of the buffer zones are shown in Table 3. The density of disaster point at different levels of each factor is shown in Figure 3h.

##### Precipitation

Rainfall is usually considered as the most vital evaluation factor for the susceptibility evaluation of geological disasters. According to statistical analysis, most landslides in Nujiang Prefecture are rainstorm-type landslides. The influence of rainfall on geological disasters is mainly due to the following aspects: water erosion on the surface leading to soil erosion, a large amount of rainwater infiltration, softening rock strata, increasing slope weight, decreasing slope stability, and induced slope slip. The Kriging interpolation method based on the GIS spatial analysis function was used to perform spatial interpolation of the rainfall data of 11 meteorological stations in Nujiang Prefecture. The natural breakpoint method was used to divide the rainfall data into five grades (Table 3), and the classification of annual average precipitation in Nujiang Prefecture was obtained (Figure 2i). The density of disaster point at different levels of each factor(Figure 3i).

##### NDVI

Normalized difference vegetation index (NDVI) was first proposed by Rouse et al. in the 1970s [46,47,48]. Because NDVI can better reflect vegetation growth and vegetation coverage, it has been used by numerous scholars. The larger the NDVI value, the larger the vegetation coverage [44,47,48]. In the present study, the MODIS data for 2019 with a spatial resolution of 250 m was selected, the vegetation coverage was extracted, and the outliers were removed using the ENVI5.3 software. The NDVI values were converted to 0–1 by using the fuzzy membership tool in the ArcGIS overlay analysis tool. The natural breakpoint method was used to divide normalized NDVI values into five grades (Figure 2j), and the CF values within the range of different NDVI grades were determined (Table 3).The density of disaster point at different levels of each factor is shown in Figure 3j. NDVI reflects the quantitative relationship between landslide disaster and vegetation density. NDVI can be calculated by the near-infrared band IR and infrared band R obtained from satellite images, as follows:(3)$$\mathrm{NDVI}=\;\frac{\mathrm{IR}-R}{\mathrm{IR}+R}$$

In Equation (5), the value range of NDVI is −0.26 to 0.80, and it is divided into five levels by using the equal spacing method: which are <0, 0–0.2, 0.2–0.4, 0.4–0.6, and >0.6.

The results show that except for interference areas such as water bodies and clouds, there is a trend that the better the vegetation coverage, the larger the CF value. This is contrary to our common conclusion, but it is not a calculation error; instead, it is because Nujiang Prefecture is a typical landform of the southwest alpine valley area with high mountains and steep slopes, poor soil, underdeveloped root systems, and unprotected vegetation. When the vegetation is destroyed, the impact force generated causes the slope to move and displace, thereby increasing the susceptibility to geological disasters. Therefore, in the southwest alpine and valley area, good vegetation coverage is not necessarily conducive for reducing the occurrence of geological disasters. On the contrary, in areas with larger slopes, the greater the NDVI value, the more likely the occurrence of geological disasters.

#### 2.3.4. Sampling Strategy of Modeling Samples

Before modeling, the positive and negative samples in the study area must be sampled. The positive sample is the landslide point in the study area, and the negative sample is the non-landslide point. The selection of the negative sample is very important for the construction of the model. Because the specific location of the landslide-prone area cannot be accurately determined before the prediction, selection of non-landslide points in the landslide-prone area must be avoided; this will help maintain the prediction accuracy of the model. Therefore, in this study, the CF value of each factor was calculated first, and then, the sum of the CF values of all the factors under each grid was calculated to obtain the susceptibility index based on the CF model. A quick evaluation was performed, and finally, by using the natural breakpoint method, landslide susceptibility was divided into five grades: low-prone area, less-prone area, medium-prone area, higher-prone area, and highly prone area. By using the CF model as the a priori model, non-landslide points were randomly selected in areas other than the high-prone area so as to ensure the accuracy of the selection of non-landslide points considering the uncertainty and spatial correlation of landslide-prone area. A total of 561 high probability non-landslide points were selected in the study area. The combination of the existing 561 landslide points and the 561 high-probability non-landslide points selected using the CF model was used as the training and test datasets for the modeling. Among them, 70% of the data was used as the training set and 30% as a test set.The process is shown in Figure 4.

### 2.4. Model Construction and Application

The CF value of each factor calculated using the CF model was used as the classification data of the SVM model. By using the selected classification data, the appropriate parameters were selected, and the model was trained. Finally, the trained model was used to perform predictions for the entire study area, and the evaluation results of disaster susceptibility were obtained.

After the modeling samples were selected, the model for the study area was constructed. The overall idea of evaluating the susceptibility of the study area is as follows. The CF formula was used to calculate the susceptibility of each factor as the classification data of the SVM model. Then, the selected classification data were used to select appropriate parameters to train the model. Finally, the trained model was used to predict landslide susceptibility for the entire study area.

Using GIS as a platform, the CF value of each index factor was calculated under different state grading by using the CF model. Next, the resolution of the 10 factor layers was unified to 30 m, and the CF values of the factors were added with equal weights. Finally, The landslide susceptibility index of Nujiang Prefecture was reclassified using ArcGIS. As shown in Figure 5, Nujiang Prefecture was divided into areas with five levels of susceptibility: extremely high susceptibility, high susceptibility, medium susceptibility, low susceptibility, and very low susceptibility.

ArcGIS was used to extract the extremely low-prone areas and low-prone areas in the CF model results. Then, 561 non-landslide points and 561 landslide points in the extremely low- and low-prone areas were selected, and the landslide and non-landslide unit spatial data were obtained. Next, the data were normalized. The spatial data of landslide and non-landslide units were divided into 70% and 30% for the training set and test set, respectively. The model was developed using SPSS Modeler18; then, the data were inputted into the SVM model for training and testing. To study the accuracy of the model, four kernel functions, namely linear kernel function, polynomial kernel function, radial basis kernel function, and Sigmoid kernel function, were used for training and testing. The one yielding the highest accuracy was selected: radial basis kernel function. Next, the spatial normalized data for the grid unit of Nujiang Prefecture were inputted into the trained model to obtain the Nujiang Prefecture landslide susceptibility index. Finally, ArcGIS was used to reclassify the landslide susceptibility index, as shown in Figure 6. Accordingly, Nujiang Prefecture was divided into areas with five levels of susceptibility: extremely high susceptibility, high susceptibility, medium susceptibility, low susceptibility, and very low susceptibility.

The four kernel functions are as follows:

(1) Linear kernel function:(4)$$K\;\left(x,x_{i}\right)=x_{i}*x\;$$

(2) Polynomial kernel function:(5)$$K\;\left(x,x_{i}\right)={\left(x_{i}*x+1\right)}^{p}$$

(3) Radial basis kernel function:(6)$$K\;\left(x,x_{i}\right)=\exp\;\left(-\gamma {\left|\left|x_{i}-x\right|\right|}^{2}\right)$$

(4) Sigmoid kernel function:(7)$$K\;\left(x,x_{i}\right)=\tan [v\;\left(x_{i}*x\right)+c]$$

## 3. Results

### 3.1. Factor Importance

The importance of index factors reflects the influence degree of different index factors on regional landslide susceptibility. Therefore, calculating and analyzing the importance of each index factor can provide a guiding basis for landslide disaster management. The predictive ability of the indicator factors used in this study is shown in Figure 7. The results obtained by the CF-SVM model show that the elevation (0.301) of landslide adjusting factors is the highest, followed by land use (0.289), proximity to road (0.171), proximity to fault (0.072), proximity to rivers (0.061), NDVI (0.034), slope (0.021), precipitation (0.020), aspect (0.019), and lithology (0.012).

### 3.2. Landslide Susceptibility Maps

Through ArcGIS10.6 software, the trained model was used to calculate the landslide susceptibility index (LSI), as shown in Figure 6. It can be seen from Figure 6 that the probability of landslide in the whole study area is −1~1. The CF susceptibility index ranged from −0.6997 to 0.4867. The prevalence index of SVM ranged from −0.7236 to 0.6788. The susceptibility index of CF-SVM ranged from −0.9104 to 0.7543. Using the natural discontinuity point classification method in ArcGIS, LSI value is divided into five easy levels: extremely high, high, medium, low, and extremely low, as shown in Figure 7. According to Figure 7, extremely high, high, medium, low, and extremely low levels of CF accounted for 10.03%, 19.60%, 26.02%, 27.24%, and 17.12%, respectively. SVM was 7.67%, 18.22%, 26.29%, 28.00%, and 19.82%; CF-SVM accounted for 7.09%, 16.57%, 10.10%, 30.11%, and 35.63%, respectively.

#### 3.2.1. Evaluation Results of Susceptibility Based on CF Model

The CF value of each index factor under different state grading was calculated and then inputted into each factor layer; then, the ArcGIS raster calculator was used for overlapping with equal weights to obtain the Nujiang prefecture susceptibility index map. The results show that the extremely high- and high-prone areas cover 4355.67 km^2^, which is only 25.89% of the total area, but include 532 landslide points, accounting for approximately 94.83% of the total number of geological disasters, and the density of disaster points is as high as 0.1221/km^2^, which is extremely low. The proportion of disaster points in low-prone areas is only 0.36%.

#### 3.2.2. The Susceptibility Evaluation Results Based on the SVM Model

The spatial normalized data of the grid unit in Nujiang Prefecture was inputted into the trained SVM model to obtain the susceptibility index of the grid unit in Nujiang Prefecture. The natural breakpoint method was then used to determine the landslide susceptibility of the grid unit in Nujiang Prefecture. The evaluation index was reclassified to obtain the susceptibility index map of Nujiang Prefecture. The results show that the extremely high- and high-prone areas cover 3806.92 km^2^, accounting for only 23.66% of the total area, but include 472 landslide points, accounting for approximately 84.13% of the total number of geological disasters, and the density of disaster points is as high as 0.1239/km^2^, which is extremely low. Further, the proportion of disaster points in low-prone areas is only 1.25%.

#### 3.2.3. Evaluation Results of Susceptibility Based on the Coupling of CF and SVM

The proportion of disasters in high-risk areas can reflect the scientific nature of model evaluation. It is more convenient for government departments to include more disaster units in high-risk areas. In this study, the GIS field calculator was used to determine the area and proportion of qualitative disaster susceptibility grades as well as the number of geological disaster points in each grade and their proportions and density (Table 4, Table 5 and Table 6). The disaster densities in the extremely high- and high-prone areas of CF, SVM, and CF + SVM are 0.1221, 0.1239, and 0.1351 disasters/km^2^, respectively. The results show that the extremely high- and high-risk areas evaluated by CF + SVM have a higher proportion of landslide disaster, which is more suitable for the practical application of landslide susceptibility in Nujiang Prefecture. The CF + SVM model performed better than the individual CF and SVM models. The results obtained using the CF + SVM model showed that the areas with extremely high vulnerability to geological disasters in Nujiang Prefecture are mainly distributed along the banks of the Dulong River, Nujiang River, Lancang River, and their tributaries as well as along roads at all levels. The analysis result is consistent with the actual geological hazard distribution characteristics in the study area. The extremely high- and high-prone areas cover 3479.05 km^2^, accounting for only 23.66% of the total area, but include 470 landslide points, accounting for 83.77% of the total number of geological disasters, and the density of disaster points is as high as 0.1351/km^2^, indicating the susceptibility to geological disasters. The higher the number, the greater the probability of the total number of geological disasters. The CF and SVM models also showed similar effects. In summary, Nujiang Prefecture should strengthen the prevention and control of geological disasters in extremely high- and high-prone areas in the future.

### 3.3. Test and Comparison of Models

Area under curve (AUC) is defined as the area under the ROC curve; the value ranges between 0.5 and 1. The larger the AUC value, the higher the prediction accuracy of the model. Figure 8 shows that the AUC values of the CF, SVM, and CF + SVM models are 0.865, 0.892, and 0.925, respectively, indicating that the three models have high accuracy and that the coupled model yields higher accuracy than the individual models. Thus, the coupling of CF and SVM models is more suitable for the assessment of landslide susceptibility.

## 4. Discussion

Most of the historical slope geological disaster sites in Nujiang Prefecture are located in the highly susceptible areas, mainly distributed along rivers and roads in Nujiang Prefecture. The results are in good agreement with the actual occurrence of slope geological disasters, indicating that the selected sensitivity factors and evaluation models are reasonable. These results are consistent with those reported in other studies assessing landslide susceptibility [39,40,42,49,50,51].

Currently, landslide researchers have applied various machine learning methods to different areas with different results. Even within a single region, different models, such as logistic regression and support vector machines, may produce different results due to weighted differences, which in turn are related to their probability distribution functions. These differences stem in part from the choice of model and uncertainty in the input data. Today, many works focus only on the application of a single model to susceptibility assessment (e.g., [52,53,54,55,56]). In this study, statistical models and machine learning methods are coupled. The SVM algorithm is enhanced by creating, using, and testing an integrated CF-SVM model. The results show that the proposed model provides higher prediction accuracy than SVM algorithm. It performs better than a single model. Based on the training and validation datasets, the model successfully distinguishes the landslide-prone areas in the study area. Our results support previous studies showing that coupled models can significantly reduce overfitting and noise problems in the modeling process [13,23,54,57,58,59,60,61,62]. The novelty of our method is that we consider the combination of statistical models and machine learning models, which can perform well in solving the problem of poor performance of a single model.

To improve land management and distribution policies, it is essential to designate landslide-prone areas. Machine learning algorithms are widely used in landslide susceptibility mapping. The main objective is to study and capture the nonlinear relationship between landslide events and their conditional parameters. However, there are still some shortcomings: (1) The selection and analysis of evaluation factors are insufficient. Due to the complex geological structure of the study area, there may be obvious correlation between landslide disaster influencing factors, or the control factors and influencing factors of screening factors are insufficient, which may lead to the decrease of model accuracy. (2) Insufficient normalization of evaluation factors: There are often differences in attributes and dimensions between influencing factors and controlling factors, which will lead to the loss of important disaster evaluation factors. (3) The sample set construction is insufficient; the negative sample selection method is especially very important. Whether the data set cleaning is in place, negative sample selection rules, and the imbalance of positive and negative sample ratio will directly affect the evaluation accuracy of the model. The coupling method of CF and SVM proposed in this paper selects factors after multivariate collinearity diagnosis, emphasizing the importance of data cleaning. Quantization and normalization of the impact factors, complete deletion of null values, replacement of noise values, interpolation outliers and other cleaning work for the original data set, and standard selection of negative samples can ensure that each feature has the same impact on the evaluation results so as to ensure the accuracy of the results. This research method effectively solves the above problems and improves the evaluation accuracy of landslide susceptibility.

The occurrence of landslide is a complex process involving geology, mechanics, meteorology and hydrology, cartography, and other fields of knowledge. Only using the model to evaluate the landslide susceptibility can only roughly predict the range of landslide occurrence, but it is not very accurate. In addition, the number of models selected in this paper is limited, and multi-model optimization comparison can be further carried out to obtain more accurate results. Due to the accelerated urbanization and intense human engineering activities in Nujiang Prefecture, China, in recent years, it is recommended to conduct a vulnerability evaluation every three years with sufficient data on landslide disaster sites and compare the evaluation results to provide a basis for local disaster prevention and reduction and reasonable land planning.

## 5. Conclusions

Taking Nujiang Prefecture as the study area and by analyzing the data, 10 evaluation factors were selected. Landslide susceptibility was evaluated using CF, SVM, deterministic coefficient, and SVM coupling (CF-SVM). The following conclusions were drawn:

(1) Landslides are common in China’s Nujiang Prefecture, where they cause severe damage to roads, buildings, and other infrastructure. Moreover, future losses are likely to grow as the economy grows. The Chinese government and departments at all levels in Nujiang Prefecture are concerned about the possible loss of life caused by the landslide. To address this issue, Chinese policymakers and policymakers need to better understand where landslides are likely to occur. The accurate landslide sensitivity map provided in this paper can help them select suitable sites for infrastructure development.

(2) In this paper, the prediction accuracy of support vector machine and CF model and their combination in the study area is obtained. The AUC of CF, SVM, and CF-SVM models were 0.865, 0.892, and 0.925, respectively, indicating that the prediction accuracy of the CF-SVM model was the highest, and the model was more suitable for landslide hazard susceptibility evaluation in the study area. This study solves the problem that a single model cannot effectively evaluate the susceptibility of landslide disaster and provides a new idea for the study of landslide disaster in Nujiang Prefecture. It provides an important decision-making basis for disaster prevention and reduction, territorial space planning, and dynamic monitoring in Nujiang Prefecture.

## Figures and Tables

**Figure 1 ijerph-19-14248-f001:**
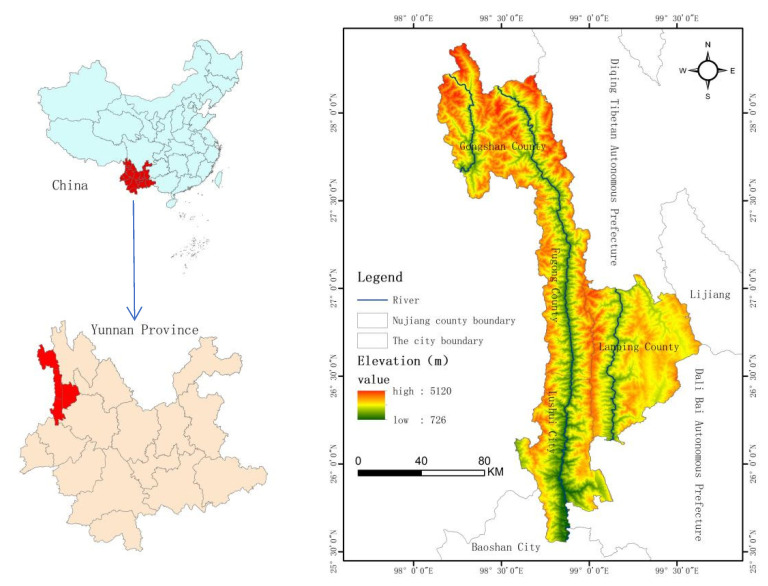
Location map of the study area.

**Figure 2 ijerph-19-14248-f002:**
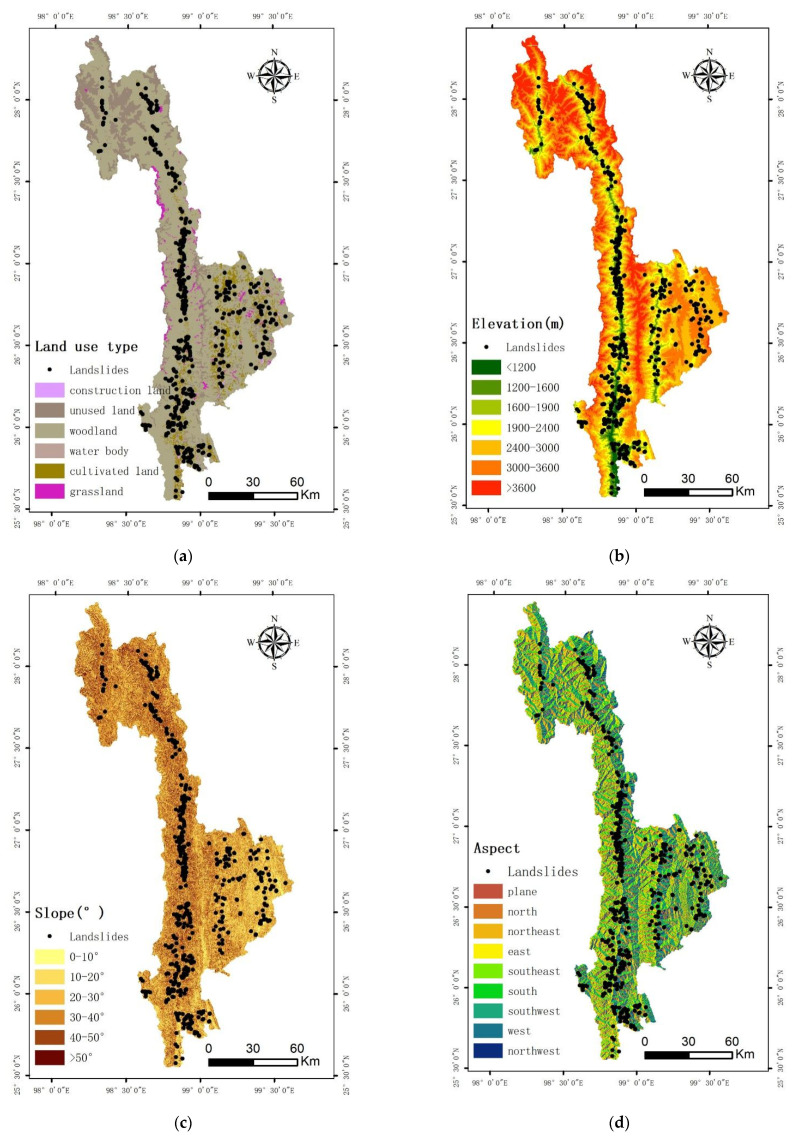
Single-factor grading diagram: (**a**) land-use type; (**b**) elevation; (**c**) slope; (**d**) aspect; (**e**) proximity to rivers; (**f**) lithology; (**g**) proximity to faults; (**h**) proximity to road; (**i**) precipitation; (**j**) NDVI.

**Figure 3 ijerph-19-14248-f003:**
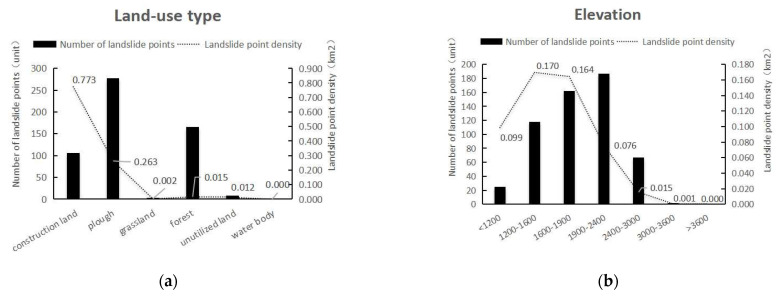
The density of disaster point at different levels of each factor: (**a**) land-use type; (**b**) elevation; (**c**) slope; (**d**) aspect; (**e**) proximity to rivers; (**f**) lithology; (**g**) proximity to faults; (**h**) proximity to road; (**i**) precipitation; (**j**) NDVI.

**Figure 4 ijerph-19-14248-f004:**
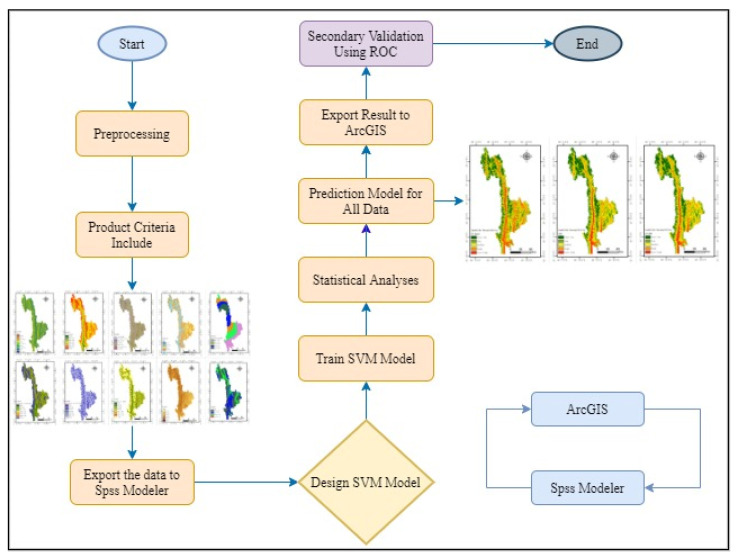
Graphical description of the methodological approach pursued CF-SVM-based LSM.

**Figure 5 ijerph-19-14248-f005:**
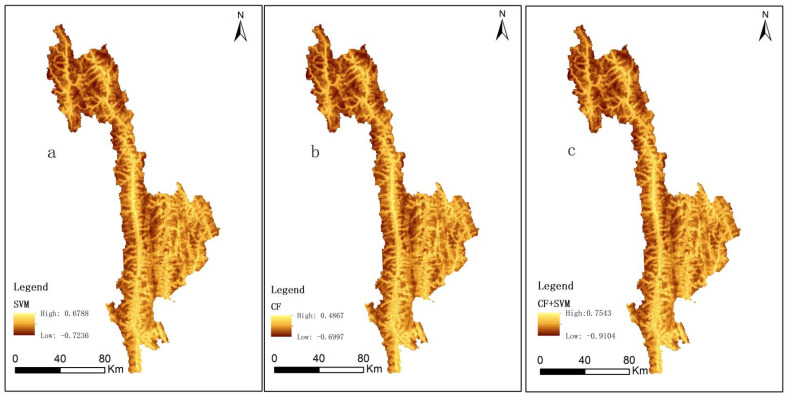
Slide susceptibility index diagram.

**Figure 6 ijerph-19-14248-f006:**
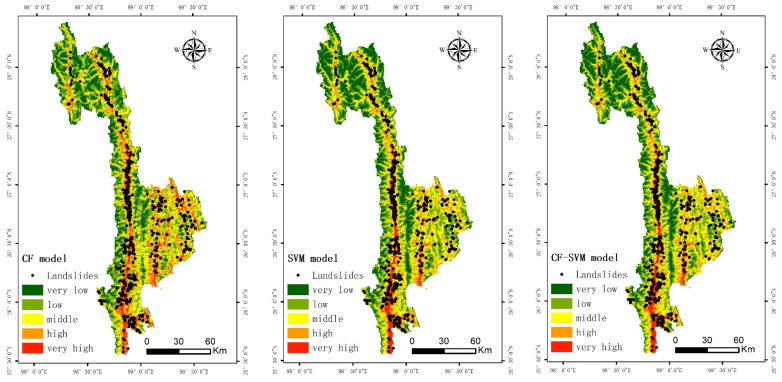
Evaluation grade of landslide susceptibility of each model.

**Figure 7 ijerph-19-14248-f007:**
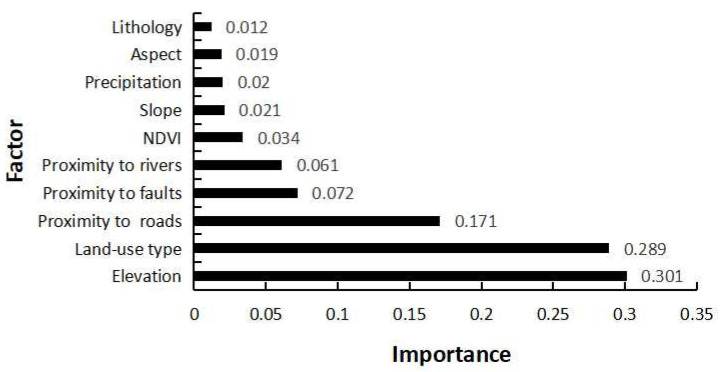
Factor importance.

**Figure 8 ijerph-19-14248-f008:**
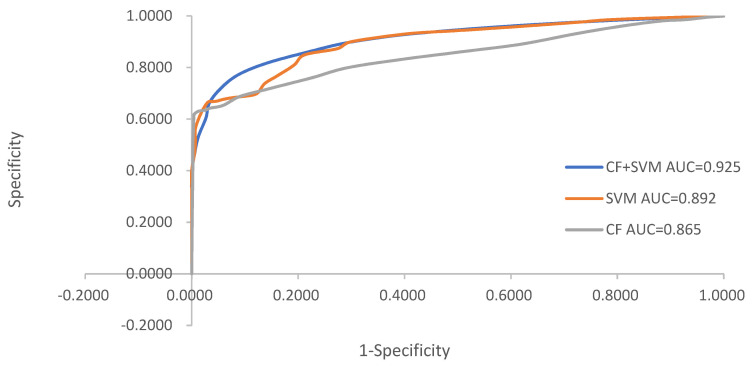
ROC curve.

**Table 1 ijerph-19-14248-t001:** Source and scales for the landslide conditioning factors used in this study.

Conditioning Factor	Source	Scale	Classification Method
Elevation	DEM was derived from ASTER GDEM data of the Geospatial Data Cloud (http://www.gscloud.cn/ (accessed on 25 January 2022).)	30 × 30 m	Manual
Aspect			Manual
Slope			Manual
Lithology	Geological map provided by Nujiang State Land and Land Bureau.	1:250,000	Lithological units
Proximity to faults			Equal interval
Proximity to rivers	The National Basic Geographic Information Database (https://wwwngcc.cn/ (accessed on 12 November 2021).)	--	Equal interval
Proximity to road		--	Equal interval
NDVI	The normalized difference vegetation index (NDVI) data were obtained from NASA (https://www.nasa.gov/ (accessed on 5 November 2021).)	250 × 250 m	Natural breaks
Precipitation	The meteorological data were procured from the Nujiang Meteorological Bureau and Water Bureau.	--	Natural breaks
Land-use type	The land-use type data were carried out based on the Landsat 8 OLI/TIRS data of Geospatial Data Cloud (http://www.gscloud.cn/ (accessed on 25 December 2020).)	The interpretation accuracy reached 94.17%.	Land-cover unit

**Table 2 ijerph-19-14248-t002:** Correlation values of each factor.

Factor	Elevation	Aspect	Slope	Lithology	Proximity to Faults	Proximity to Rivers	Proximity to Road	NDVI	Precipitation	Land-Use Type	Agrotype
Elevation	1.000										
Aspect	0.070	1.000									
Slope	−0.095	−0.046	1.000								
Lithology	0.413	0.078	−0.037	1.000							
Proximity to faults	0.054	0.031	−0.019	0.170	1.000						
Proximity to rivers	0.074	0.078	0.036	0.109	−0.059	1.000					
Proximity to road	0.139	0.092	0.007	0.109	0.063	0.034	1.000				
NDVI	0.216	0.048	−0.132	0.092	−0.045	−0.001	−0.008	1.000			
Precipitation	−0.160	−0.138	0.082	−0.309	0.026	0.105	−0.066	−0.077	1.000		
Land-use type	−0.065	−0.009	0.193	−0.052	−0.020	0.128	0.032	−0.107	0.092	1.000	
Agrotype	0.371	−0.009	−0.105	0.065	0.250	0.131	0.048	0.184	0.068	0.121	1.000

**Table 3 ijerph-19-14248-t003:** CF values of each factor.

Factor	Classification	Area	Number of Landslide Points	pa	ps	CF
Slope	0–10°	731.0079	30	0.0410	0.0382	0.0731
	10–20°	2240.5428	116	0.0518	0.0382	0.2735
	20–30°	3820.0815	187	0.0490	0.0382	0.2293
	30–40°	4425.603	153	0.0346	0.0382	−0.0973
	40–50°	2680.7157	63	0.0235	0.0382	−0.3933
	>50°	805.0491	12	0.0149	0.0382	−0.6186
Aspect	Plane	22.2255	0	0.0000	0.0382	−1.0000
	North	1886.0004	40	0.0212	0.0382	−0.4538
	Northeast	1702.0764	68	0.0400	0.0382	0.0467
	East	1852.425	100	0.0540	0.0382	0.3048
	Southeast	1806.8571	75	0.0415	0.0382	0.0840
	South	1907.3826	56	0.0294	0.0382	−0.2375
	Southwest	1849.5159	66	0.0357	0.0382	−0.0671
	West	1862.0028	96	0.0516	0.0382	0.2703
	Northwest	1814.5143	60	0.0331	0.0382	−0.1379
Elevation	<1200	253.4256	25	0.0986	0.0382	0.6375
	1200–1600	695.9151	118	0.1696	0.0382	0.8057
	1600–1900	985.4766	162	0.1644	0.0382	0.7984
	1900–2400	2467.7622	187	0.0758	0.0382	0.5162
	2400–3000	4342.7025	67	0.0154	0.0382	−0.6050
	3000–3600	3874.6638	2	0.0005	0.0382	−0.9870
	>3600	2083.0542	0	0.0000	0.0382	−1.0000
Proximity to rivers	0–200	1610.7471	94	0.0584	0.0382	0.3599
	200–400	1539.6921	119	0.0773	0.0382	0.5264
	400–600	1476.4599	84	0.0569	0.0382	0.3424
	600–800	1412.6364	90	0.0637	0.0382	0.4170
	800–1000	1348.4016	66	0.0489	0.0382	0.2292
	1000–1200	1273.5279	51	0.0400	0.0382	0.0491
	>1200	6041.535	57	0.0094	0.0382	−0.7599
Lithology	Weak rock group	7073.2944	230	0.0325	0.0382	−0.1528
	Hard rock group	4066.362	148	0.0364	0.0382	−0.0479
	Harder rock group	3545.0187	183	0.0516	0.0382	0.2712
	Loose rock group	18.3249	0	0.0000	0.0382	−1.0000
Proximity to faults	<400	1764.5994	94	0.0533	0.0382	0.2950
	400–800	1569.7656	119	0.0758	0.0382	0.5164
	800–1200	1359.7416	84	0.0618	0.0382	0.3975
	1200–1600	1274.3034	90	0.0706	0.0382	0.4780
	1600–2000	999.1170	66	0.0661	0.0382	0.4392
	2000–2400	844.9938	51	0.0604	0.0382	0.3824
	>2400	6890.4792	57	0.0083	0.0382	−0.7897
NDVI	Poor vegetation cover	289.2655	0	0.0000	0.0382	−1.0000
	Average vegetation coverage	980.5989	21	0.0214	0.0382	−0.4483
	Good vegetation coverage	3216.5431	102	0.0317	0.0382	−0.1744
	Excellent vegetation cover	5367.6063	245	0.0456	0.0382	0.1706
	Very excellent vegetation cover	4832.6326	193	0.0399	0.0382	0.0464
Proximity to road	0–200	1753.1496	275	0.1569	0.0382	0.7868
	200–400	1560.8115	100	0.0641	0.0382	0.4205
	400–600	1390.5945	45	0.0324	0.0382	−0.1570
	600–800	1249.3899	33	0.0264	0.0382	−0.3161
	800–1000	1111.7763	29	0.0261	0.0382	−0.3248
	1000–1200	1087.8549	31	0.0285	0.0382	−0.2606
	1200–1400	863.1207	18	0.0209	0.0382	−0.4631
	>1400	5686.3026	30	0.0053	0.0382	−0.8663
Land-use type	Construction land	137.1537	106	0.7729	0.0382	0.9883
	Plowland	1056.7953	278	0.2631	0.0382	0.8889
	Grassland	1206.9702	3	0.0025	0.0382	−0.9372
	Forest	10918.0086	166	0.0152	0.0382	−0.6108
	Unutilized land	666.0549	8	0.0120	0.0382	−0.6935
	Water body	718.0173	0	0.0000	0.0382	−1.0000
Precipitation	852–1021	4331.9367	207	0.0478	0.0382	0.2095
	1021–1152	3233.5686	131	0.0405	0.0382	0.0605
	1152–1299	2430.3624	50	0.0206	0.0382	−0.4705
	1299–1451	2797.758	130	0.0465	0.0382	0.2284
	1451–1656	1909.3743	43	0.0225	0.0382	−0.4192

**Table 4 ijerph-19-14248-t004:** Statistical table of CF model susceptibility.

Degree of Susceptibility	Area (km^2^)	Ratio (%)	Number of Disasters (Unit)	Ratio%	Disaster Point Density (Unit/km^2^)
Very low susceptibility	2516.5000	17.12%	0.0000	0.0000	0.0000
Low susceptibility	4005.7900	27.24%	2.0000	0.36%	0.0005
Moderate susceptibility	3825.0400	26.02%	27.0000	4.81%	0.0071
High susceptibility	2881.4300	19.60%	150.0000	26.74%	0.0521
Very high susceptibility	1474.2400	10.03%	382.0000	68.09%	0.2591
Total	14,703.0000	1.0000	561.0000	1.0000	—

**Table 5 ijerph-19-14248-t005:** Statistical table of SVM model susceptibility.

Degree of Susceptibility	Area (km^2^)	Ratio (%)	Number of Disasters (Unit)	Ratio%	Disaster Point Density (Unit/km^2^)
Very low susceptibility	2914.3200	19.82%	0.0000	0.0000	0.0000
Low susceptibility	4116.1100	28.00%	7.0000	1.25%	0.0017
Moderate susceptibility	3865.6500	26.29%	82.0000	14.62%	0.0212
High susceptibility	2678.8600	18.22%	173.0000	30.84%	0.0646
Very high susceptibility	1128.0600	7.67%	299.0000	53.30%	0.2651
Total	10,473.00	1.0000	561	1.0000	—

**Table 6 ijerph-19-14248-t006:** Statistical table of CF + SVM model susceptibility.

Degree of Susceptibility	Area (km^2^)	Ratio (%)	Number of Disasters (Unit)	Ratio%	Disaster Point Density (Unit/km^2^)
Very low susceptibility	5238.37002	35.63%	0.0000	0.0000	0.0000
Low susceptibility	4427.62662	30.11%	22	3.92%	0.0049
Moderate susceptibility	1557.94512	10.10%	69	12.30%	0.0442
High susceptibility	2435.83212	16.57%1	154	27.45%	0.0632
Very high susceptibility	1043.22612	7.09%	316	56.33%	0.3029
Total	10,473.00	1.0000	561	1.0000	—

## Data Availability

Not applicable.

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
