# Peer review of "Evaluation of Landslide Susceptibility Based on CF-SVM in Nujiang Prefecture"

_ijerph, 2022, doi:10.3390/ijerph192114248_

Round 1

Reviewer 1 Report

Reviewer comments

Thank you for submitting your paper to IJERPH. I read carefully manuscript number: ijerph-1913246, the manuscript entitled: " Evaluation of landslide susceptibility based on GIS in Nujiang Prefecture ". In this study, landslide susceptibility was assessed to understand the characteristics and formation mechanism of regional landslides and provide relevant references and suggestions for national spatial planning and disaster prevention and mitigation. The support vector machine (SVM) classifier was used to evaluate the landslide susceptibility in the study area based on the grid unit, elevation, gradient, slope direction, formation lithology, fracture structure, distance from road, distance from drainage, vegetation coverage, land use type, and rainfall by using the deterministic coefficient method. In my point of view, the result of this kind of research could be interesting and useful for many applications specifically for the spatial mapping of multi-index analysis. Please check the English grammar. The English language is moderate. Please check all parts of the manuscript and correct grammatical errors. Some sections of paper require major revisions before any further. I attached my reviewer supplementary comments in the below and pdf file.

1- Abstract

1-1- The abstract section need to complete with more information. The abstract should be improved.

1-2-The concrete finding of this research need to be added to the abstract section.

2- Introduction

2-1- In the literature review section, use newer references related to landslide susceptibility assessment from 2020 to 2022.

2-2- The literature review is too general and thus can’t indicate any novelty of the current study. It is better that explain more about the novelty of manuscript in introduction section. The manuscript has not quite innovative. Please explain about its novelty.

2-3- The writing structure of the article should be improved.

2-4- Research organization not provided.

3- Research Methodology section were provided in poor way. So needs improvements

4 -"Results and Discussion" were provided in poor way.

4-1-Results of this study need to be compared with previous research works. Authors are emphatically recommended to provide a new section for this purpose.

4-2- Is it possible to evaluate the importance of each index?

5- Conclusion section need rewriting.

Author Response

Dear reviewer:   Thank you for your decision and constructive comments on my manuscript. We have carefully considered the suggestion of Reviewer and make some changes. We have tried our best to improve and made some changes in the manuscript.   The highlighted part that has been revised according to your comments. 

Reviewer 2 Report

This manuscript evaluated the landslide susceptibility in Nujiang Prefecture using SVM models. It is meaningful for disaster prevention and mitigation. Nevertheless, there are some aspects that the authors should take into account in the revision process:

(1) The models are conducted using ArcGIS and SPSS Moeller. These models are usually used in relevant models. Then, please state the breakthrough of this research clearly. The authors should to state the understanding of SVM method, the improvement of landslide vulnerability assessment and the application process of the method. At present, the paper is just a description of the principle, lacking specific innovative research work.

(2) Explain the applicability of the data source in the paper, especially the specific research work of the author in the data source. The criteria of selecting the factors. It should reference the characteristics of the study area and the disaster status. The reasons for choosing these factors should state clearly.

(3) The aim of this research should be clear. Presenting a novel model, performance comparison of three models and the evaluation of landslide susceptibility in Nujiang Prefecture, these make the aim confusing.

Additionally, there are some written problems. Some of the written problems are listed the following:

(1) The quality of Figure 2.

(2) The classification method for the factors. It states natural break point method is used. But a rounding method is used in table 2 and figure 3.

(3) The classification criteria for the degree of susceptibility for the three models. 

Author Response

(The authors gave the same response as above.)

Reviewer 3 Report

The complex terrain, unique climate and fragile ecological environment of Nujiang Prefecture lead to the frequent occurrence of geological disasters in this area. At present, there is an urgent need for landslide disaster susceptibility evaluation model suitable for Nujiang Prefecture. The support vector machine (SVM) classifier was used to evaluate the landslide susceptibility in the study area based on the grid unit, elevation, gradient, slope direction, formation lithology, fracture structure, distance from road, distance from drainage, vegetation coverage, land use type, and rainfall by using the deterministic coefficient method. The topic is interesting and within the scope of IJERPH. The main conclusions are well supported by results. I would recommend a Moderate Revision.

(1)   There are so many machine learning methods that can be used in classification. Why the authors only use the SVM method?

(2)   There are so many long sentences in the manuscript. Please avoid using long sentences.

(3)   Introduction: The landslide susceptibility may be highly related with climate change. This is a background context of such studies. The climate change issue should be briefly discussed in the manuscript.  The following references may help.

Zscheischler, J., Martius, O., Westra, S., Bevacqua, E., Raymond, C., Horton, R. M., et al. (2020). A typology of compound weather and climate events. Nature Reviews Earth & Environment, 1(7), 333–347. https://doi.org/10.1038/s43017-020-0060-z

Yin, J., Slater, L., Gu, L., Liao, Z., Guo, S., & Gentine, P. (2022). Global increases in lethal compound heat stress: Hydrological drought hazards under climate change. Geophysical Research Letters, 49, e2022GL100880

(4)   The main limitation and future works should be discussed.

(5)   The language can be polished.

Author Response

(The authors gave the same response as above.)

Reviewer 4 Report

The title of the article should be more specific.

The abstract needs to be rewritten.

The introduction is general and there is no proper literature gap to focus on. Authors should clearly point out the motivation of this work. In addition, many hybrid methods are missing, such as AHP + frequency ratio.

The quality of figures should be improved.

“Data sources”: The information of different data should be shown in a table, so that readers can clearly see them, such as data source, resolution, time, etc.

A description of landslides in the study area is missing, including the type, size, source, etc. It's better to provide some photos. In addition, the location of landslides should be shown on the figures.

Line 112, “… coupled with CF model and SVM model ...”, CF/SVM? Such abbreviations in the manuscript must be defined at their first mention.

It’s better to analyze the importance of such evaluation factors and remove irrelevant factors. Please refer to 10.1016/j.catena.2020.104851 and 10.1007/s10346-019-01286-5

The spelling or grammatical errors need to be checked carefully.

Author Response

(The authors gave the same response as above.)

Round 2

Reviewer 1 Report

Thank you for submitting your revised paper to IJERPH. I read carefully manuscript number: ijerph-1913246, the manuscript entitled: " Evaluation of landslide susceptibility based on CF-SVM in Nujiang Prefecture ". In my point of view, the result of this kind of research could be interesting and useful for many applications specifically for the spatial and temporal landslide susceptibility, inundation and environmental risk mapping of multi-index analysis. All previous comments were applied. The authors applied all comments point by point and I confirm their revision. The added information is important and useful and led to improving the manuscript. I accept the revised manuscript in this present form. I concur; the final decision is accepted for publication.

Author Response

Dear Reviewers,

Thanks very much for taking your time to review this manuscript. I really appreciate all your comments and suggestions! Your advice has been of great help to us. Thank you for your affirmation of our research, which gives us great motivation to carry out landslide related research.

Reviewer 2 Report

The authors have made corresponding revisions according to or comments. However, there are some aspects that the authors should deal with further.

(1) In “Response to Reviewer 2 Comments” file, the authors state that “The main purpose of this paper is to propose a new model (CF-SVM)”. The CF-SVM coupling model has been widely used in relevant research, although it is perhaps the first time in Nujiang landslides. So based on the purpose of this manuscript, I do not think this research has important innovations or breakthroughs.

(2) The authors have stated the SVM model specifically in the revised manuscript, whereas these models are conducted using SPSS Modeler software. The specific innovative work is still scarce.

Hence, the manuscript should focus on its innovations and state them clearly. I do not think the current state is enough. I suggest to thoroughly and completely discuss the independent innovation research work of the author of CF-SVM, what specific improvements and optimizations have been made, and whether it is innovative.

Author Response

Dear Reviewers,

Thanks very much for taking your time to review this manuscript. I really appreciate all your comments and suggestions! Please find my itemized responses in below and my revisions in the re-submitted files.

Reviewer 4 Report

Please check the paper carefully to make sure there are no errors

Author Response

Dear Reviewers,

Thanks very much for taking your time to review this manuscript. I really appreciate all your comments and suggestions! Please find my itemized responses in below and my revisions in the re-submitted files.

Point 1:  Please check the paper carefully to make sure there are no errors.

Response 1: Thank you for your suggestion. Thank you for giving us good advice, which has greatly improved our articles. We've been working on the manuscript for so long that we've combed through the text and had native English speakers make language corrections.